# Experimental and Numerical Study on the Mechanical Performance of Ultra-High-Performance Concrete T-Section Beams

**Jianluan Li [1,2], Yonggao Yin [1,*] and Jing Yan [1,*]**

1   College of Civil Engineering, Hefei University of Technology, Hefei 230009, China; jluanli@163.com
2   Anhui Transport Consulting and Design Institute Co., Ltd., Hefei 230088, China
*   Correspondence: 2021800035@hfut.edu.cn (Y.Y.); 2020110640@mail.hfut.edu.cn (J.Y.)

**Abstract:** Aiming to investigate the mechanical performance of UHPC T-section beams, five specimens are fabricated and tested, considering the variable steel fiber volume fraction (SFVF). The code of the Association Francaise de Génie Civil (AFGC) is evaluated by test data. Additionally, based on Abaqus (2020), refined finite element analysis (FEA) models of specimens are established and validated by experimental data. Moreover, the parametric sensitivity analysis is carried out, which aims to further investigate the effect of shear span ratio, longitude reinforcement ratio, and stirrup ratio on the bending-shear behavior of T-section beams. The test results indicated that the ultimate load of the specimen improves with the increase of SFVF, and the use of steel fibers can greatly improve the shear capacity instead of the bending capacity. Furthermore, SFVF can change the failure mode; the specimens fail in shear failure when SFVF < 2%, while they fail in bending failure when SFVF ≥ 2%. From the evaluation of codes, the AFGC code is conservative in the prediction of ultimate capacity, which can guide the design of UHPC structures well. Additionally, from the parametric analysis of FEM, the failure mode transformed from shear failure to bending failure as the shear span ratio increased, particularly in specimens with SFVF ≥ 2.5%. Moreover, the stirrup ratio $\rho_{sv}$ has a significant effect on the shear performance of structures with SFVF ≤ 1%, while it has less effect with SFVF ≥ 2%.

**Keywords:** ultra-high-performance concrete (UHPC); T-section beam; four-point loading test; ultimate capacity; finite element analysis (FEA)

## 1. Introduction

Ultra-high-performance concrete (UHPC) has been widely utilized in precast girders as a new material to reduce structural self-weight and improve construction efficiency [1]. In conventional concrete structures, stirrups are required to resist the shear forces generated by external loads and self-weight. However, for UHPC structures, owing to the addition of steel fibers and the small cross-section, it is difficult to construct UHPC structures with dense stirrups [2]. Furthermore, the tensile and compressive strengths of UHPC are greatly improved with the addition of steel fibers; thus, to replace the effect of stirrups, it is reasonable to use dense and randomly distributed steel fibers of UHPC in the structure for resisting shear force [3–5]. UHPC beams with steel fibers are more reasonable in terms of shear force than conventionally reinforced concrete beams.

Influential factors affecting the bending-shear performance of UHPC beams are similar to those of conventional concrete beams, such as shear span ratio, longitudinal reinforcement ratio, stirrup ratio, and material strength. To gain a thoughtful understanding of UHPC beams, numerous studies [6–13] have been conducted to investigate the structural behavior of UHPC beams. Based on the test, Yang, I.H., et al. [6] found that the addition of SFVF up to 1.5% can increase the load capacity, ductility, and bending rigidity of the UHPC beams, whereas SFVF = 2.0% did not significantly increase the ductility or bending



rigidity. In addition, to research the effect of fiber properties (i.e., fiber type, geometry, dosage, orientation, etc.) on the mechanical performance of UHPC, a series of tests were conducted in numerous studies [7,14–18].

Moreover, the size effect also has a large impact on the shear performance of beams. Matta [11] concluded that the shear strength of large-size beams was reduced by an average of 55% compared to small specimens by testing beams without web reinforcements at different section sizes. Along similar lines, Bentz [19] studied the effect of section height on GFRP-reinforced concrete beams and found that the section height affects the size effect in the range of 30–50%. At present, many scholars and codes have proposed prediction formulas for the flexural and shear capacities of UHPC beams. Following this, to evaluate the existing analytical approaches when calculating the moment capacity of UHPC beams, Shafieifar [20] found that ACI544 [21] and FHWA [22] methods can predict the ultimate moment capacity of UHPC beams with acceptable accuracy. Likewise, Zhang et al. [8] have put forward the predictive equations for the ultimate flexural capacity of T-section UHPC beams and validated them with experimental results. Furthermore, to study the shear strength of a T-section beam without stirrups, Voo et al. [10] proposed a calculation model based on crack sliding and plasticity theory. In terms of structural failure mode, bending failure is a ductile failure; thus, the beam can have a large non-linear deformation with less reduction in strength and stiffness when bending failure occurs. While shear failure is a brittle failure, shear failure of the beam is often accompanied by a large degradation of stiffness and strength, and the damage is sudden and has a greater impact on structural safety [2,23–26]. Therefore, aim to make the structure have a good deformation capacity and energy dissipation capacity under dynamic load. Most structural design codes, such as AFGC codes [27], follow the structural design principles of strong shear capacity and weak bending capacity. There have been some similar T-beam tests in previous studies [2–11], and many useful conclusions have been drawn. Generally, most of the past studies mainly focused on how to enhance the mechanical performance of UHPC, but systematic studies on the shear-bending properties of UHPC beams with different SFVF are limited. Especially the UHPC design codes needed to be evaluated with experimental data. Nevertheless, to promote the application of UHPC while decreasing the cost, it is necessary to further optimize the mix ratio of UHPC.

Based on the above consideration and discussion, to investigate the mechanical performance of UHPC T-section beams without stirrups, five specimens were designed and tested, considering the variable steel fiber volume fraction (SFVF). Following that, the code of the Association Francaise de Génie Civil (AFGC) [27] is evaluated by test data. Additionally, the refined finite element analysis (FEA) models were established based on Abaqus (2020), which have been validated by experimental data. Finally, the parametric analysis is conducted to further study the effect of shear span ratio, longitude reinforcement ratio, and stirrup ratio on the bend shear behavior of beams. The outcome of this study is intended to provide some guidance for better performance evaluation and design of UHPC bridges.

## 2. Test Program

### 2.1. Specimen Design

To investigate the effect of steel fiber contents on the cracking pattern and failure of UHPC T-section beams, five specimens were constructed. The basic dimensions and test parameters are shown in Figure 1 and Table 1. In detail, the shear span ratio $\lambda$, reinforcement ratio, and stirrup ratio of the test specimens are 2.2%, 7.0%, and 0%, respectively. The T-section beam has a length of 1.5 m and a height of 0.2 m. HRB400 steel is adopted for longitudinal reinforcement with a standard yield strength of 400 MPa, which is in accordance with China codes [28], whereas the diameter is 20 mm. The thickness of the concrete protective layer $c$ is 20 mm. It should be noted that specimen T-1 is cast with C50 normal concrete, which aims to form a comparison. Furthermore, the UHPC used in this study was formulated by combining cement, microsilica, fine sand, steel

fibers, water, and a water-reducing admixture. The bridging mechanism of steel fibers can significantly improve the strength of concrete materials [29–32]. Subsequently, the straight steel fibers (see Figure 2) were adopted with a length, diameter, and modulus of 13 mm, 0.12 mm, and 210 GPa, respectively, and their tensile strength was greater than 2850 MPa. In detail, Figure 2 depicts the schematic diagram of the steel fibers and the specimens casting progress.

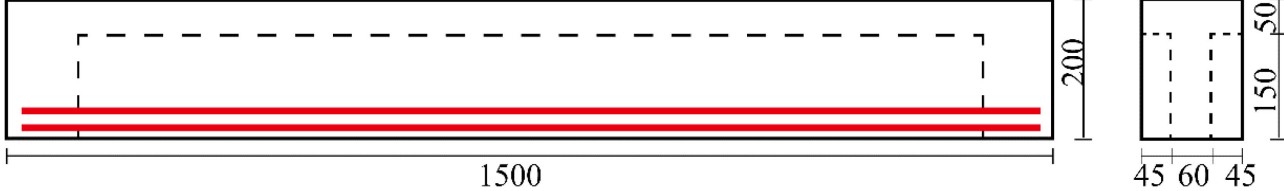

**Figure 1.** 2D diagram of push-off test specimens (unit: mm).

**Table 1.** Design of specimens for push-off tests.

| Specimens | SFVF (%) | Shear Span Ratio (λ) | Longitudinal Reinforcements | Reinforcement Ratio (%) | Stirrup | Stirrup Ratio (%) |
|---|---|---|---|---|---|---|
| T-1 | 0 (NC) | 2.2 | 2Φ20 | 7.0 | 0 | 0 |
| T-2 | 1.0% | 2.2 | 2Φ20 | 7.0 | 0 | 0 |
| T-3 | 2.0% | 2.2 | 2Φ20 | 7.0 | 0 | 0 |
| T-4 | 2.5% | 2.2 | 2Φ20 | 7.0 | 0 | 0 |
| T-5 | 3.0% | 2.2 | 2Φ20 | 7.0 | 0 | 0 |

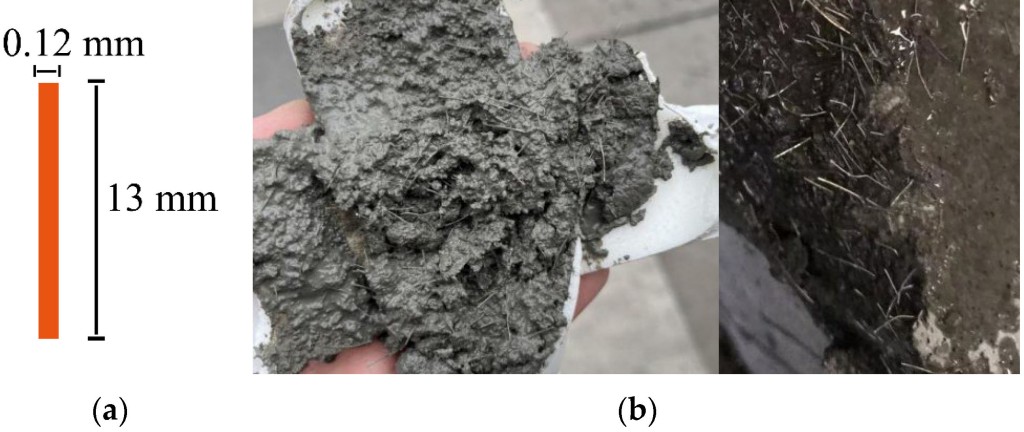

(**a**)                    (**b**)

**Figure 2.** Steel fibers: (**a**) schematic diagram of the steel fibers and (**b**) sampling detection.

Figure 3 presents the setup of the tests. The specimen was simply supported at both ends, and a two-point vertical load was applied in the middle of the specimen with a point-to-point distance of 608 mm. Such a loading method divided the specimen into three regions along the specimen length, including two shear-bending regions and a pure bending region in the middle. In the pure bending region, strain gauges were installed on the midspan to measure the strain distribution along the beam height. In the shear-bending regions, strain gauges were attached diagonally to measure the strain responses under combined shear and bending forces. Additionally, displacement gauges were installed at the bottom of the specimen to monitor the deflections of the specimen.

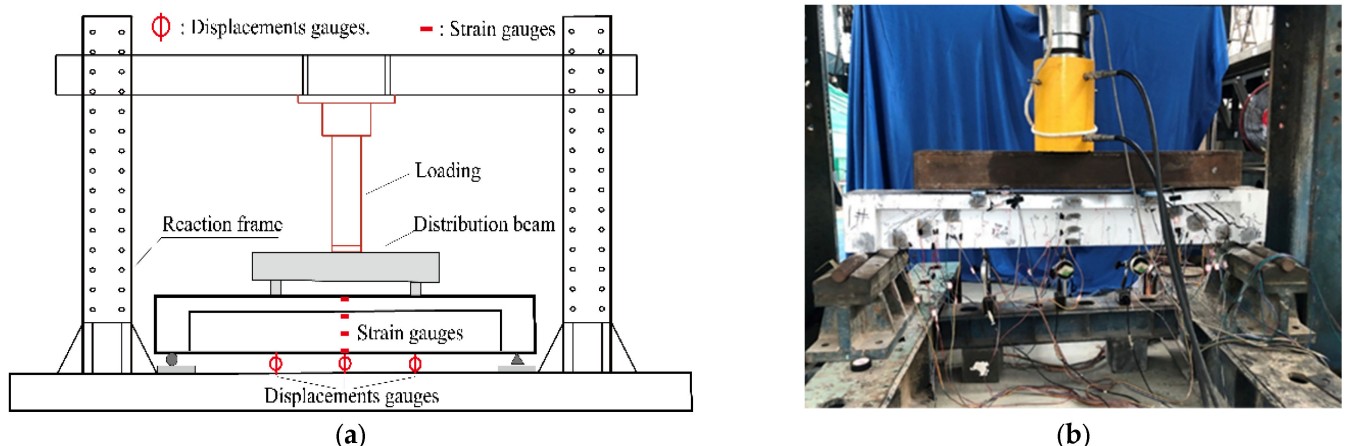

**Figure 3.** Test set-up: (**a**) loading apparatus and (**b**) test photo.

*2.2. Test Results*

Failure modes and crack patterns of specimens are shown in Figure 4. Load-midspan displacements are illustrated in Figure 5. As can be seen from Figures 5 and 6, the steel fiber content has a big impact on the failure mode of specimens. Based on the experimental phenomenon (see Figure 4), T-1 and T-2 are shear failures, while T-3~5 are bending failures. That is, the specimen fails in shear failure when SFVF is lower than 1%. However, when the fiber admixture is higher than 1%, the specimen fails due to bending failure. Notably, the shear failure of the beam is brittle, while the bending failure has good ductility.

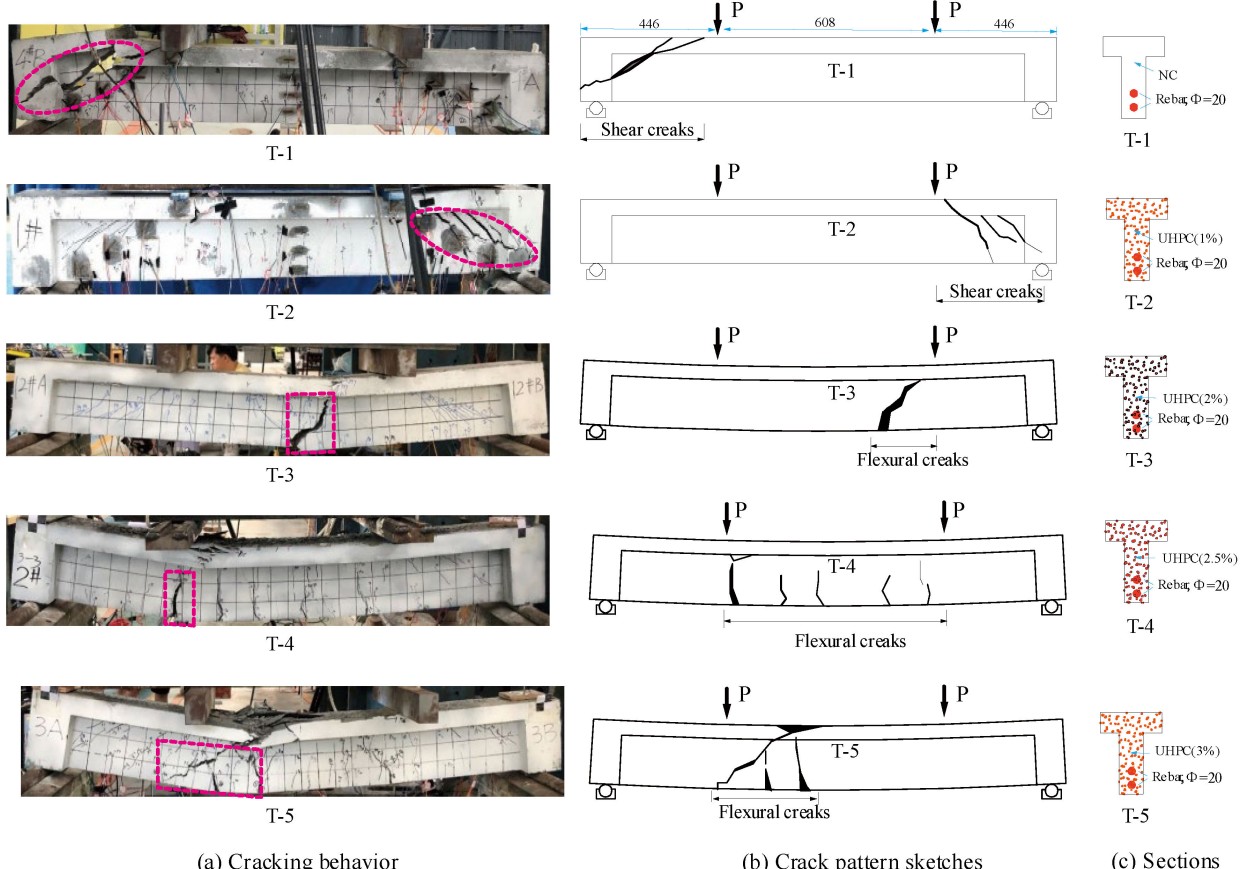

**Figure 4.** Failure modes and crack patterns of different SFVFs.

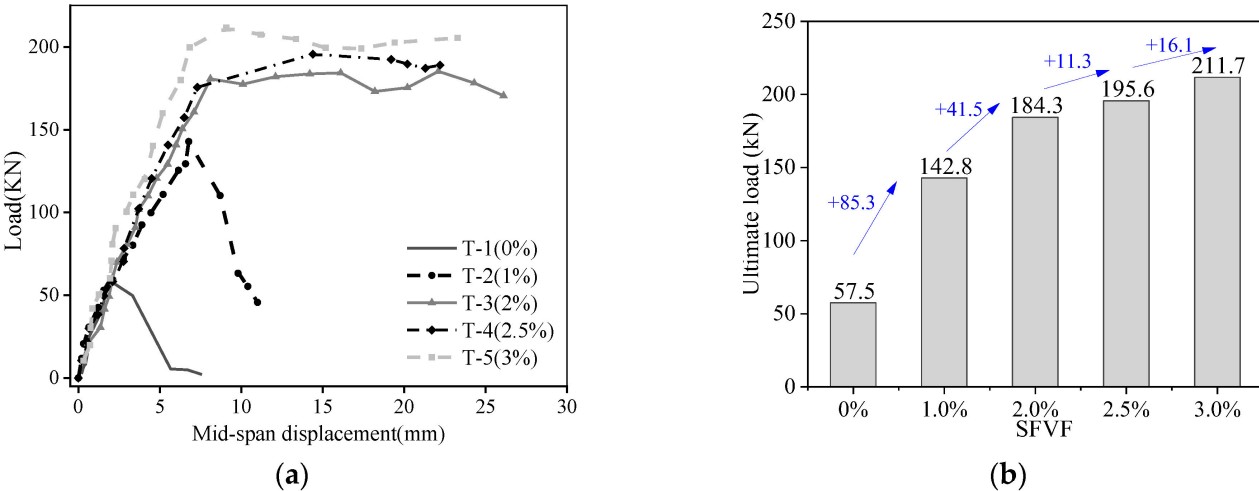

**Figure 5.** Test results of specimens: (**a**) load-displacement curves and (**b**) ultimate load histogram.

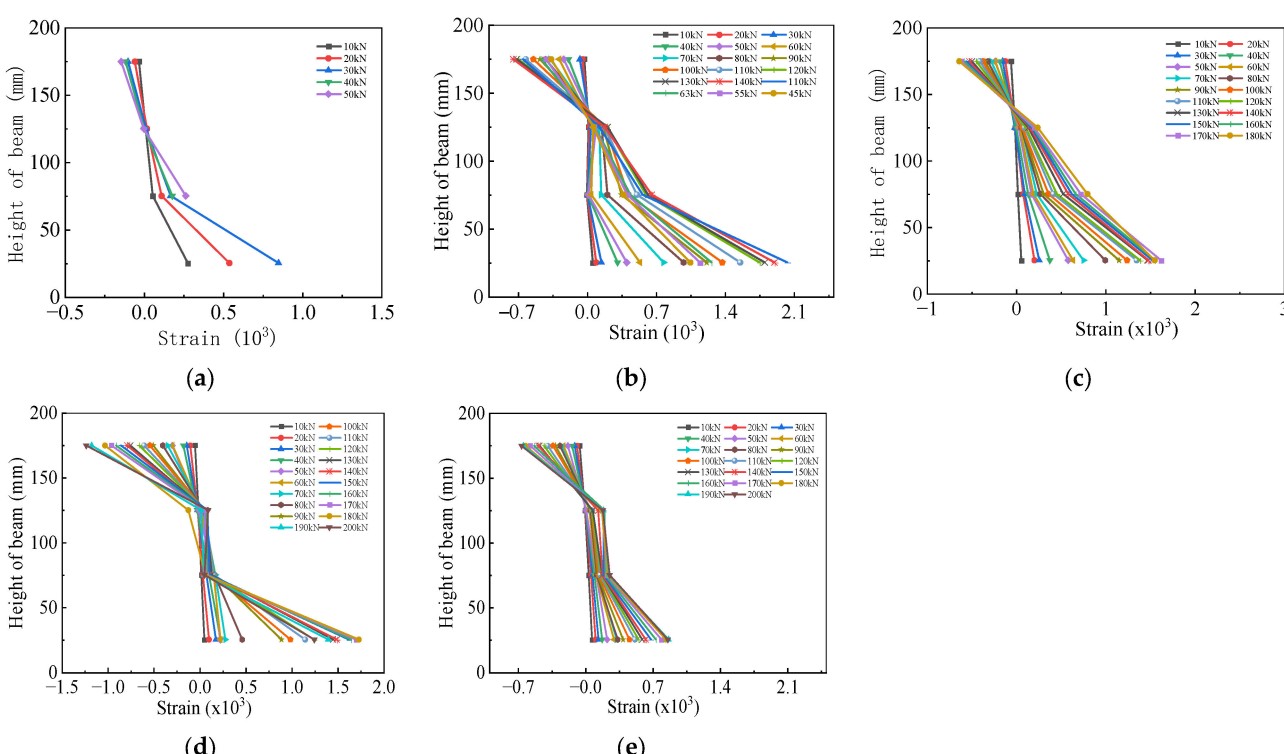

**Figure 6.** Strain distribution of the midspan section: (**a**) T-1 (SFVF = 0%); (**b**) T-2 (SFVF = 1%); (**c**) T-3 (SFVF = 2%); (**d**) T-4 (SFVF = 2.5%); and (**e**) T-5 (SFVF = 3%).

From Figure 5, the ultimate loads of specimens T-1~5 are 57.5 kN, 142.8 kN, 184.3 kN, 195.6 kN, and 211.7 kN, respectively. Compared with T-1, the ultimate load of T-2~5 increased by 149.7%, 220.5%, 240.2%, and 268.2%, respectively. Therefore, it can be seen that the ultimate load of the specimen improves with the increase in steel fiber content. The ultimate capacity of T-2~5 (with steel fibers) compared with that of T-1 (no steel fibers) and T-2~5 (with steel fibers) shows that using steel fibers can significantly improve the shear capacity of the beam. Moreover, the bearing capacity is not directly proportional to the steel fiber content. When SFVF is greater than 2.0%, the bearing capacity grows slowly with the increase in SFVF. The increase in steel fiber content is more advantageous in improving the shear capacity of the beam than the flexural capacity. According to the design principles of conventionally reinforced concrete structures, the bending capacity is greatly affected

by longitudinal reinforcement. The shear capacity is greatly affected by stirrups rather than longitudinal reinforcement. Similarly, for reinforced UHPC structures, SFVF has a greater impact on the shear capacity, and SFVF has a better effect on compressive strength improvement than tensile strength [2,27]. Consequently, in this study, since the specimen has a high longitudinal reinforcement ratio without stirrups, it can be concluded from test results that SFVF has a greater impact on the shear capacity than the bending capacity. Notably, this conclusion is in accordance with UHPC design codes [27].

Figure 6 plots the strain distribution of the midspan section of specimens. For specimen T-1 (see Figure 6a), the mid-span section remains in an elastic state, and there is obvious tensile and compressive partitioning. From Figure 6b–e, in the loading initial phase (e.g., low load), a clear assumption of flat cross-section is exhibited in specimen T-3, and the distribution of tensile and compressive zones is obvious. Moreover, it can be seen that the variation of SFVF and the strain of the midspan section exhibit little difference. In general, with the increase in SFVF, the change in strain in the tensile zone at the beam bottom is obvious. Moreover, the strain value measured at the same measurement point for each specimen always decreases with the increase in SFVF when the external load is the same.

### 2.3. Evaluation of Design Codes

To further guide engineering design, comparisons between experimental results and design codes were carried out in this section. As the first country to design and construct UHPC structures, France has a mature design specification for UHPC structures. Thus, the French UHPFRC standard (AFGC) [27] was chosen to predict the capacity of specimens. In AFGC codes [27], the calculation of shear capacity of UHPC structures is based on the truss structure model, which is mainly composed of three parts (see Equations (1)–(4)), namely, the shear capacity provided by UHPC concrete ($V_{Rd,c}$), the shear capacity provided by shear reinforcement ($V_{Rd,s}$), and the shear capacity provided by steel fiber ($V_{Rd,f}$).

$$V_{Rd} = V_{Rd,c} + V_{Rd,s} + V_{Rd,f} \tag{1}$$

$$V_{Rd,c} = \frac{0.24}{\gamma_{cf}\gamma_E} \cdot k \cdot f_{ck}^{0.5} \cdot b \cdot z \tag{2}$$

$$V_{Rd,s} = \frac{A_{sw}}{s} \cdot z \cdot f_{ywd} \cdot (\cot\theta + \cot\alpha)\sin\alpha \tag{3}$$

$$V_{Rd,f} = \frac{A_{fv} \cdot \sigma_{Rd,f}}{\tan\theta} \tag{4}$$

In which $\gamma_{cf}$, $\gamma_E$ are the safety factor; $f_{ck}$ is the characteristic value of UHPC compressive strength; and b is the rib width of the T-beam. $A_{fv}$ is the area of fiber effect. Notably, fiber effect refers to the phenomenon that fibers increase the shear capacity of structures, which is often considered in the area of the beam web based on AFGC codes. $\sigma_{Rd,f}$ is the residual tensile strength of a fiber-reinforced section; $\theta$ is the included angle between the main compressive stress and the beam axis, which is recommended to be taken as 30~40°. is the included angle between the main compressive stress and the beam axis, which is recommended to be taken as 30~40°. $k_N$ is the prestress enhancement factor, calculated $k_N$ is 1.33. $f_{ywd}$ is the tensile design value of shear reinforcement on the section.

Based on Equation (2), the shear capacity ($V_{Rd}$) of the UHPC T-section beam was obtained. Of note, the ultimate load ($F_{shear, failure}$) of specimens was calculated from Equation (5) when the shear failure occurred. In addition, the prediction of bending capacity is based on code Euro 1992-1-1 [33], and the bending capacity ($M_u$) of specimens was obtained. The ultimate load ($F_{bending, failure}$) of specimens was calculated from Equation (6) when bending failure occurred. The comparison between experimental data and the AFGC prediction value is illustrated in Figure 7. Compared with test data, the prediction values of bending capacity and shear capacity are small. Moreover, in terms of structural

failure mode, bending failure is a ductile failure, and the beam can have a large non-linear deformation with less reduction in strength and stiffness when bending damage occurs. Shear failure is a brittle failure; shear failure of the beam is often accompanied by a large degradation of stiffness and strength, and the damage is sudden and has a greater impact on structural safety [23,24]. Therefore, aim to make the structure have a good deformation capacity and energy dissipation capacity under impact load. Most structural design codes, such as AFGC codes, follow the structural design principles of strong shear capacity and weak bending capacity. Generally, the shear failure from the AFGC code is conservative due to the material safety factors, and the bending failure is also conservative. Therefore, the actual safety margins of UHPC beams are high and can be guaranteed based on the AFGC design code. That is, the AFGC code is conservative in the prediction of ultimate capacity, which can guide the design of UHPC structures well.

$$F_{\text{shear , failure}} = 2 \cdot V_{Rd} \tag{5}$$

$$F_{\text{bending, failure}} = 2 \cdot M_u \cdot l = 2 \cdot M_u \cdot 0.396 \tag{6}$$

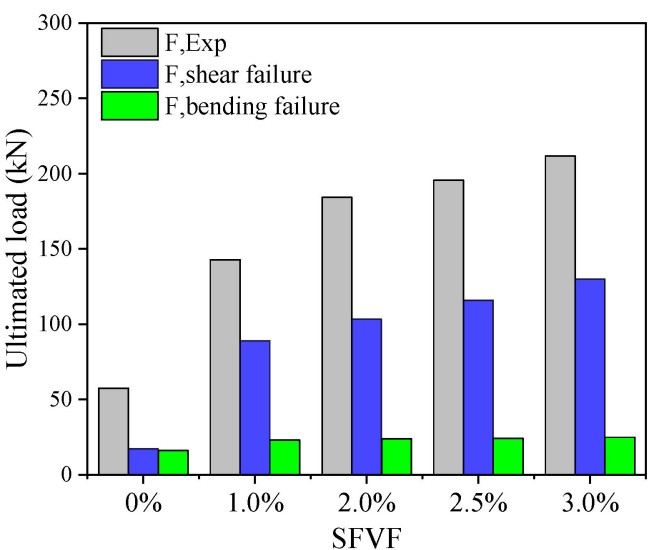

**Figure 7.** Comparisons between experimental results with AFGC codes.

## 3. Finite Element Analysis (FEA)

### 3.1. FEA Model

To gain a more thorough understanding of the shear behavior of the UHPC T-section beam, finite element modeling and analysis were conducted based on the Abaqus platform (2020) (see Figure 8). 3D finite element models of the tested specimens were established in Abaqus/Standard. The perfect elastic-plastic material model is chosen to describe the strain-stress relationship of steel bars due to its easy parameter determination and simple form (see Equation (7)). The elastic-plastic model of HRB400 steel material is illustrated in Figure 9a, whereas the yield stress and elastic modulus are 400 MPa and 190 GPa, respectively. A concrete damage plasticity (CDP) material model [16,34–36] in Abaqus was employed to simulate the stress-strain constitutive behavior of concrete. The compressive model of UHPC material uses the fitting formula proposed by literature [37], which is divided into ascending and descending sections, and the fitting formula is shown in Equation (8).

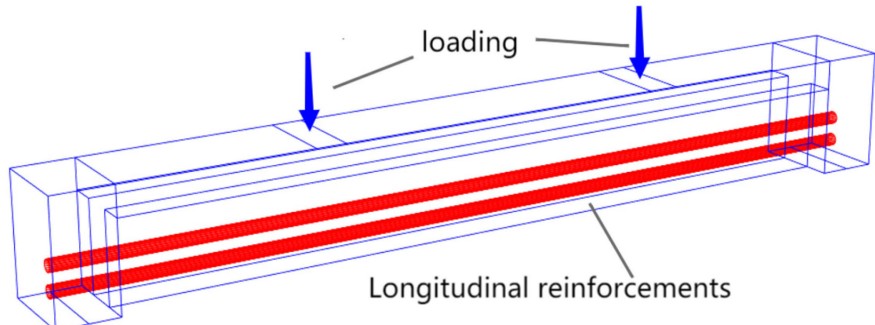

**Figure 8.** FE model diagram.

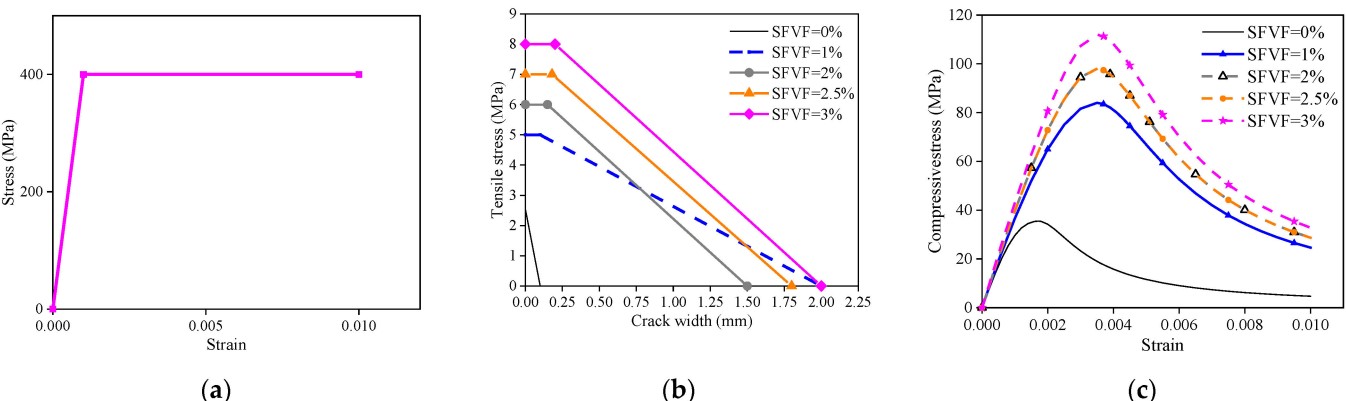

**Figure 9.** Stress-strain relationships of materials. (**a**) HRB400 steel; (**b**) tension model of UHPC; (**c**) compression model of UHPC.

In the CDP model, three models are provided to describe the uniaxial tensile performance of concrete, namely the stress-strain model, the stress-displacement model, and the fracture energy model. Since the softening stage of UHPC is mainly controlled by the crack width, the uniaxial tensile performance of UHPC is described by the stress-strain model with large errors. However, the direct input of fracture energy does not reflect the characteristics of tensile strain hardening in UHPC. Thus, the stress-displacement (crack width) model was adopted to simulate the uniaxial tensile behavior of UHPC in FEA. The stress-crack width model proposed in the literature [18] was chosen to describe the tensile properties of UHPC (Equation (9)). The C3D8R solider elements are used to model the UHPC materials, while the T3D2 truss elements are employed to simulate the behavior of steel reinforcement.

For longitudinal reinforcement,

$$\sigma_s = \begin{cases} E_s \varepsilon_s & \left(0 \leq \varepsilon_s \leq \varepsilon_y\right) \\ f_y & \left(\varepsilon_y \leq \varepsilon \leq \varepsilon_u\right) \end{cases} \tag{7}$$

In which $f_y$ is the yield stress of steel and $E_s$ is the elastic modulus.

For the compression model of concrete,

$$y = \begin{cases} ax + (6-5a)x^5 + (4a-5)x^6 & 0 \leq x \leq 1 \\ \dfrac{x}{b(x-1)^2 + x} & x \geq 1 \end{cases}$$

$$y = \sigma / f_c, x = \varepsilon / \varepsilon_0, \varepsilon_0 = 3500 \mu\varepsilon$$

$$a = \frac{E_0}{E_c}$$

$$\tag{8}$$

$E_c$ and $E_0$ are the initial elastic modulus and the secant modulus at the peak stress, respectively. $\varepsilon$ is strain.

For the tension model of concrete,

$$\sigma_t = \begin{cases} f_t & (w \le w_1) \\ \frac{f_t}{w_c - w_1}(w_c - w) & (w_1 \le w \le w_c) \end{cases} \tag{9}$$

where $\sigma_t$ is the tensile stress of UHPC, $w$ is the crack width of UHPC, $f_t$ is the tensile strength, and $w_1$ and $w_c$ are the crack widths at the end of tensile strain hardening and the ultimate crack width, respectively. In order to consider the stiffness degradation of UHPC [38], the compressive damage parameter dc proposed by Birtel et al. [39] is used in this study, as shown in Equation (10). Equation (11) is used to calculate the coefficient of tensile damage $d_t$, which is in accordance with Nie's study [23].

$$d_c = 1 - \left( \frac{\sigma_c / E}{0.2\varepsilon_{inc} + \sigma_c / E} \right) \tag{10}$$

$$d_t = 1 - \left( 1 - \frac{w}{w_c} \right)^n \tag{11}$$

In which $\varepsilon_{inc}$ is the inelastic compressive strain of UHPC (see Equation (10)), $n$ is the index of stiffness degradation, which is taken as one based on the past parametric analysis [40]. Similarly, based on the results of the parametric analysis, $w_1$ was taken as 0, 0.1, 0.15, 0.18, and 0.2 mm, and $w_c$ was taken as 0.1, 2.0, 1.5, 1.8, and 2.0 mm for UHPC with steel fiber content of 0%, 1%, 2.0%, 2.5%, and 3.0%, respectively. Detailed information about the UHPC material model is shown in Figure 9b,c and Table 2. It should be noted that the mechanical properties of SFVF = 0 were adopted in accordance with the Chinese code [28], whereas the compressive strengths of SFVF = 1%, 2%, 2.5%, and 3% were adopted based on code [41].

**Table 2.** Parameters of the material model.

|  | SFVF = 0 | SFVF = 1 | SFVF = 2 | SFVF = 2.5 | SFVF = 3 |
|---|---|---|---|---|---|
| $E_c$ | 34,500 | 40,000 | 42,000 | 42,000 | 45,000 |
| $W_1$ | 0 | 0.1 | 0.15 | 0.18 | 0.2 |
| $W_2$ | 0.1 | 2 | 1.5 | 1.8 | 2 |
| $f_t$, MPa | 2.54 | 5 | 6 | 7 | 8 |
| $f_c$, MPa | 32.4 | 84 | 98 | 100 | 112 |

The mesh size of the element has a big influence on the FE results. The smaller the mesh size, the closer the FE results are to the real values, but the computational efficiency of the model will be reduced. In consideration of calculation accuracy and efficiency, the mesh size of UHPC was adopted as 10 mm × 10 mm × 10 mm. Figure 10 shows the mesh diagrams of FEA.

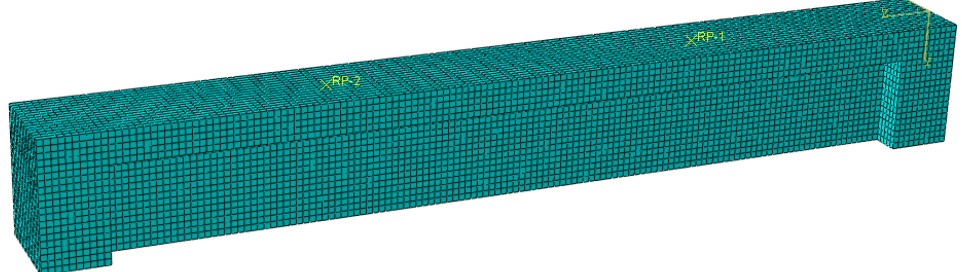

**Figure 10.** Mesh diagrams.

### 3.2. Model Validation

To verify the accuracy of FEM, the damage modes and load-deflection curves of UHPC beams obtained from FEA were compared with the experimental results. Figure 11 illustrates the experimental and numerical load-displacement responses for tested specimens T-1~5, which demonstrates that the numerical results matched well with the experimental results, particularly for the peak loads. The errors of peak loads for T-1, T-2, T-3, and T-4 specimens were 13.0%, 0.1%, 0.2%, 0.7%, and 3.1%, respectively. In the FEA, the effect of bond slip between concrete and reinforcement is not considered due to the use of embedding constraints, which makes the stiffness of the FEM curve slightly greater than that of the experiment. The average value (Av) and standard deviation (SD) of the ratio of ultimate load between FEM and test were 1.020 and 0.063, respectively (see Figure 11f). This validates the effectiveness and correctness of the adopted FE model of the UHPC T-section beam.

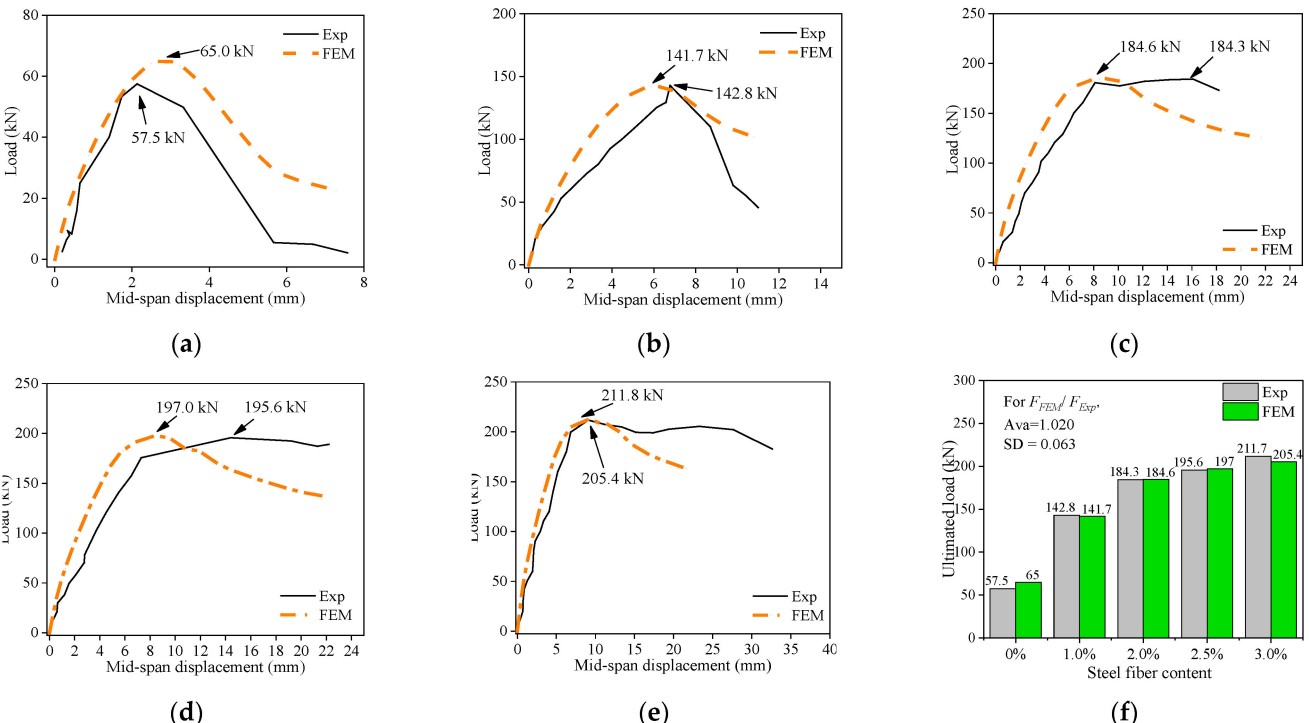

**Figure 11.** Comparisons of the experimental and simulated results: (**a**) T-1 (SFVF = 0%); (**b**) T-2 (SFVF = 1%); (**c**) T-3 (SFVF = 2%); (**d**) T-4 (SFVF = 2.5%); (**e**) T-5 (SFVF = 3%); and (**f**) ultimate load histogram.

According to the test results, the typical shear failure mode occurred in all the UHPC beams when the steel fiber content was less than 2%, which was characterized by the steel fiber being pulled out from the beam along the main diagonal crack, and the diagonal tensile damage occurred in the web of the beam, with the diagonal crack extending upward to the loading point and downward to the vicinity of the bearing. Taking specimen T-1 as an example, the shear failure mode obtained from the FEM and the test are compared, as shown in Figure 12. It can be seen that the damage characteristics of the FE beam are in good agreement with the experimental phenomenon.

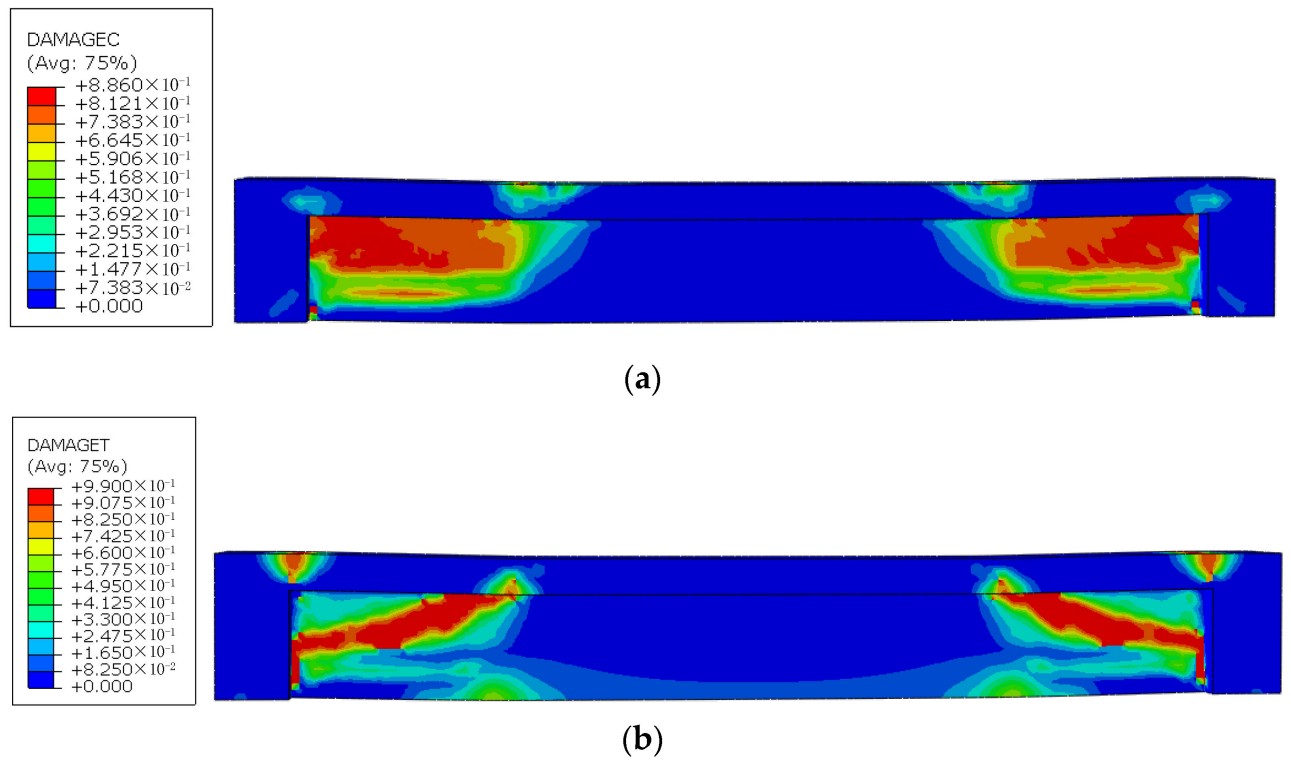

**Figure 12.** Damage contour of T-1 specimens (shear failure): (**a**) compression damage; (**b**) tension damage.

## 4. Parametric Sensitivity Analysis

To further reveal the shear performance of UHPC beams, parametric sensitivity analyses were carried out in this section based on a validated FEA model that considered the shear span ratio and reinforcement ratio. Since the mechanical properties of UHPC without steel fibers are similar to those of normal concrete, this section does not discuss the specimens without steel fiber admixture, mainly considering SFVF of 1%, 2%, 2.5%, and 3%.

### 4.1. Shear Span Ratio

The shear span ratio $\lambda$ is one of the main factors affecting the shear performance of concrete beams. To investigate the effect of shear span ratio on the shear performance of UHPC beams with different steel fiber content, a total of 12 FE models were established, and the values of $\lambda$ are 1.0, 2.0, and 3.0, respectively. Detailed information about 12 FE models is listed in Table 3, and the load-displacement curves of FE results are illustrated in Figure 13.

As indicated in Figure 13, with the increase of $\lambda$, the stiffness of the beam in the elastic phase and the bearing capacity of the specimen gradually decreased, but the ductility enhanced, and the failure mode changed from shear failure to bend-shear failure (see Figure 14). Take specimens SFVF1-$\lambda$1, SFVF1-$\lambda$2, and SFVF1-$\lambda$3 as examples. The peak load of SFVF1 decreases with $\lambda$ increasing by 22.2%, 33.9%, and 36.9%, respectively. Likewise, the peak load of SFVF2 decreases with $\lambda$ increasing by 22.2%, 33.9%, and 36.9%, respectively.

**Table 3.** Information on FE models with different shear span ratios.

| Specimen IDs | SFVF (%) | Shear Span Ratio (λ) | Longitudinal Reinforcements | Reinforcement Ratio (%) |
|---|---|---|---|---|
| SFVF1-λ1 | 1.0 | 1 | 2Φ20 | 7.0 |
| SFVF1-λ2 | 1.0 | 2 | 2Φ20 | 7.0 |
| SFVF1-λ3 | 1.0 | 3 | 2Φ20 | 7.0 |
| SFVF2-λ1 | 2.0 | 1 | 2Φ20 | 7.0 |
| SFVF2-λ2 | 2.0 | 2 | 2Φ20 | 7.0 |
| SFVF2-λ3 | 2.0 | 3 | 2Φ20 | 7.0 |
| SFVF2.5-λ1 | 2.5 | 1 | 2Φ20 | 7.0 |
| SFVF2.5-λ2 | 2.5 | 2 | 2Φ20 | 7.0 |
| SFVF2.5-λ3 | 2.5 | 3 | 2Φ20 | 7.0 |
| SFVF3-λ1 | 3.0 | 1 | 2Φ20 | 7.0 |
| SFVF3-λ2 | 3.0 | 2 | 2Φ20 | 7.0 |
| SFVF3-λ3 | 3.0 | 3 | 2Φ20 | 7.0 |

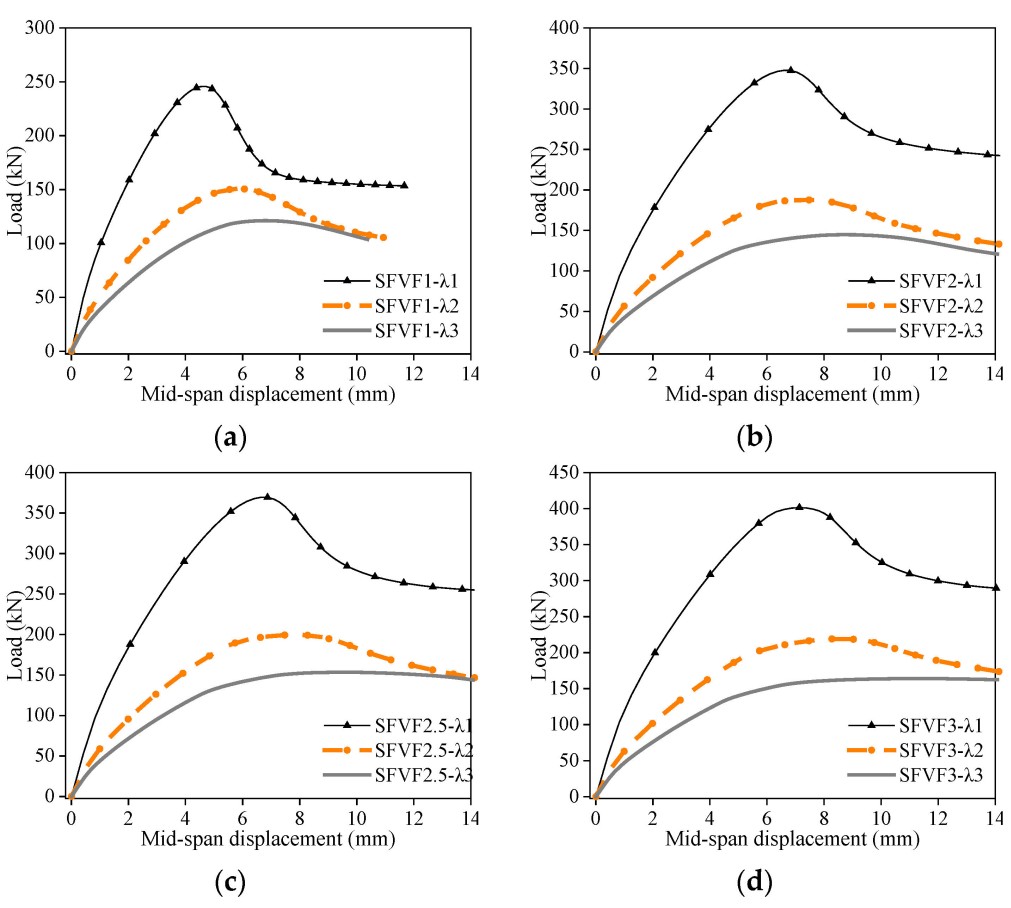

**Figure 13.** Load-mid-span displacements curves: (**a**) SFVF = 1%; (**b**) SFVF = 2%; (**c**) SFVF = 2.5%; and (**d**) SFVF = 3%.

### 4.2. Reinforcement Ratios

Also, the longitudinal reinforcement ratio $\rho$ is an important factor affecting the mechanical performance of concrete beams. To investigate the effect of longitudinal reinforcement ratio $\rho$ on the shear performance of UHPC beams with different steel fiber content, a total of 16 FE models were established, and the values of $\rho$ are 7.0%, 5.6%, 4.5%, and 3.4%, respectively. Detailed information about those FE models is listed in Table 4, and the load-displacement curves of the FE results are illustrated in Figure 15.

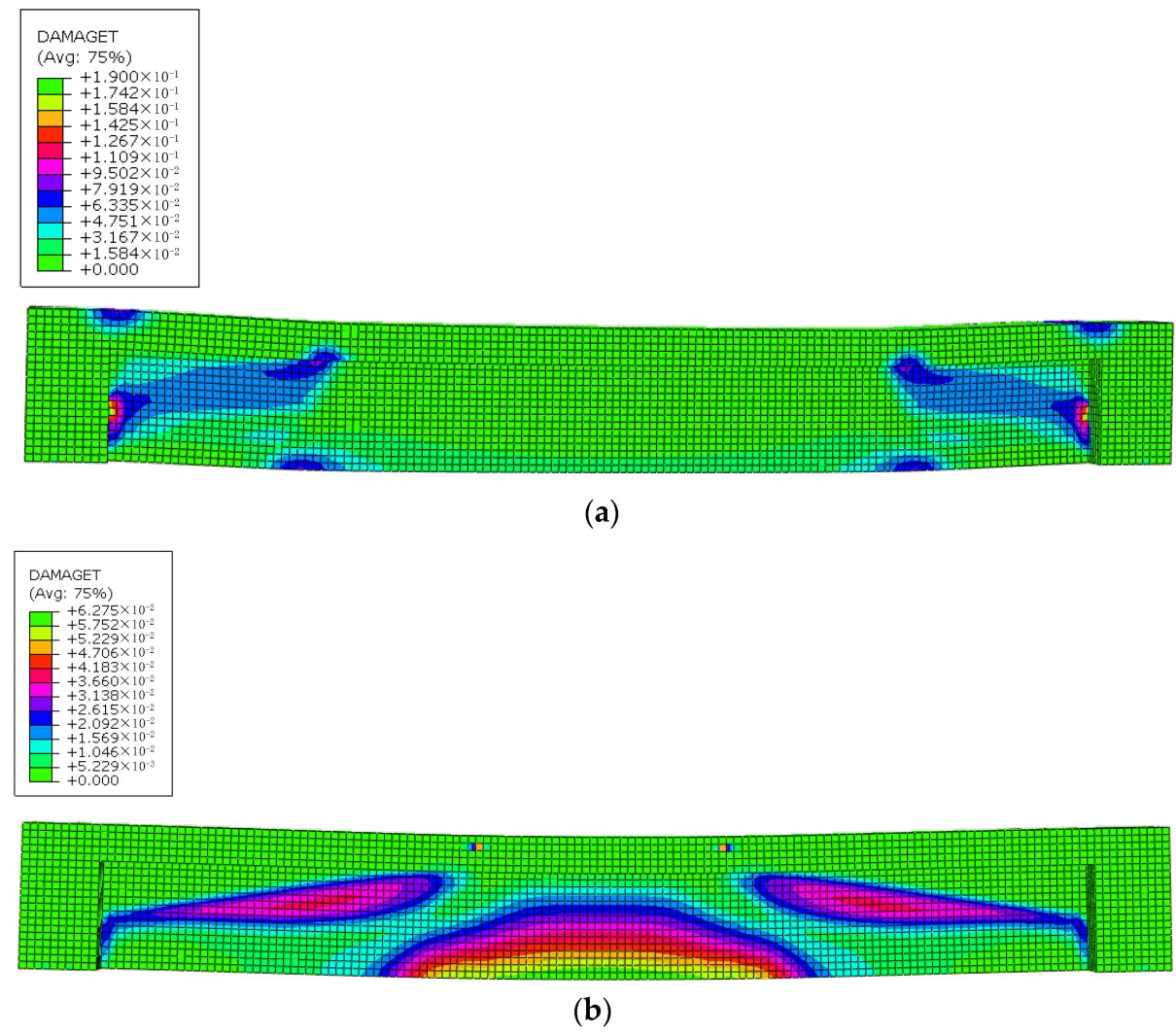

**Figure 14.** Tension damage contour of SFVF3 specimens: (**a**) SFVF3-λ2 (shear failure) and (**b**) SFVF3-λ3 (bend-shear failure).

**Table 4.** Information on FE models with different reinforcement ratios.

| Specimen IDs | SFVF (%) | Shear Span Ratio (λ) | Longitudinal Reinforcements | Reinforcement Ratio (%) |
|---|---|---|---|---|
| SFVF1-R20 | 1.0 | 2.2 | 2Φ20 | 7.0 |
| SFVF1-R18 | 1.0 | 2.2 | 2Φ18 | 5.6 |
| SFVF1-R16 | 1.0 | 2.2 | 2Φ16 | 4.5 |
| SFVF1-R14 | 1.0 | 2.2 | 2Φ14 | 3.4 |
| SFVF2-R20 | 2.0 | 2.2 | 2Φ20 | 7.0 |
| SFVF2-R18 | 2.0 | 2.2 | 2Φ18 | 5.6 |
| SFVF2-R16 | 2.0 | 2.2 | 2Φ16 | 4.5 |
| SFVF2-R14 | 2.0 | 2.2 | 2Φ14 | 3.4 |
| SFVF2.5-R20 | 2.5 | 2.2 | 2Φ20 | 7.0 |
| SFVF2.5-R18 | 2.5 | 2.2 | 2Φ18 | 5.6 |
| SFVF2.5-R16 | 2.5 | 2.2 | 2Φ16 | 4.5 |
| SFVF2.5-R14 | 2.5 | 2.2 | 2Φ14 | 3.4 |
| SFVF3-R20 | 3.0 | 2.2 | 2Φ20 | 7.0 |
| SFVF3-R18 | 3.0 | 2.2 | 2Φ18 | 5.6 |
| SFVF3-R16 | 3.0 | 2.2 | 2Φ16 | 4.5 |
| SFVF3-R14 | 3.0 | 2.2 | 2Φ14 | 3.4 |

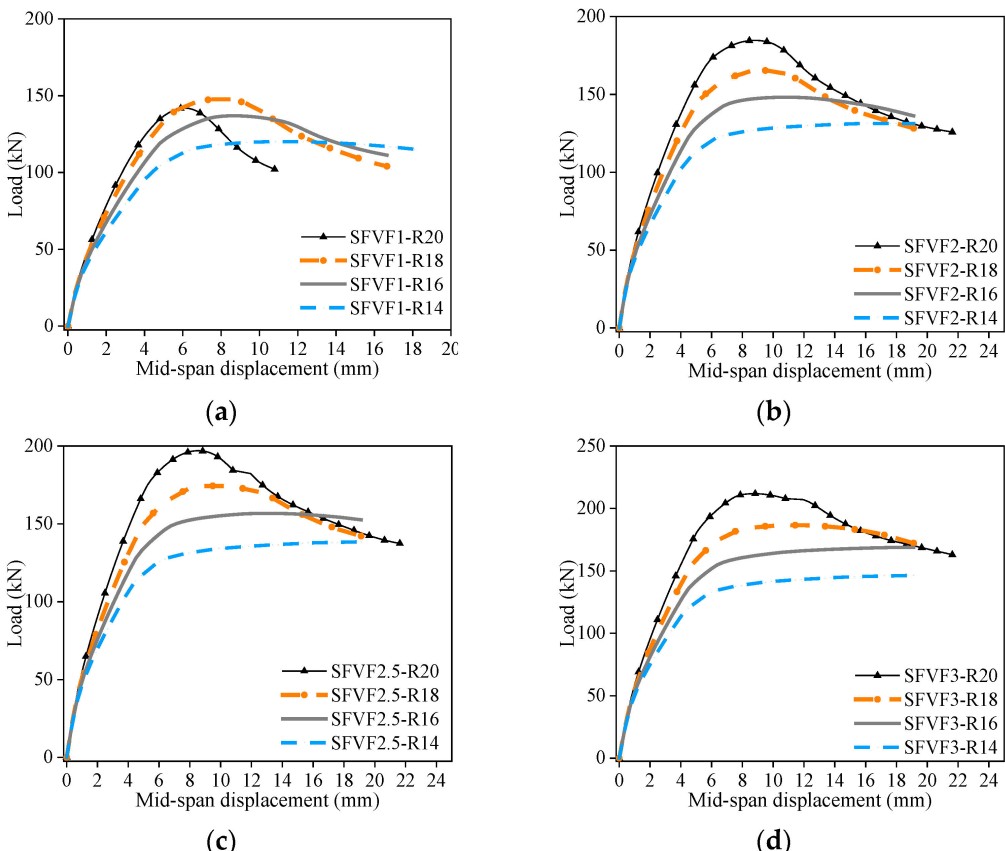

**Figure 15.** Load-mid-span displacements curves: (**a**) SFVF = 1%; (**b**) SFVF = 2%; (**c**) SFVF = 2.5%; and (**d**) SFVF = 3%.

From Figure 15, with the increase in the longitudinal reinforcement ratio, the ultimate capacity $P_u$ and initial stiffness $K_0$ of the beam increase, but the ductility decreases. Take specimens SFVF1-R14, SFVF1-R16, SFVF1-R18, and SFVF1-R20 as examples; the peak loads $P_u$ are 120.1 kN, 136.9 kN, 147.8 kN, and 141.7 kN, respectively, which indicate that the beam with SFVF = 1% has a high ultimate load capacity when the design value of ρ is 5.6% (see Figure 13a). Likewise, compared with SFVF2-R14, the peak load $P_u$ of SFVF2-R16~20 increases by 12.9%, 25.9%, and 40.5%, respectively.

*4.3. Stirrup Ratios*

The stirrup ratio $\rho_{sv}$ affects the shear performance of beams. To investigate the effect of stirrup ratio $\rho_{sv}$ on the shear performance of UHPC beams with different SFVF, a total of 12 FE models were established, and the values of $\rho_{sv}$ are 1.88%, 0.94%, and 0.62%, respectively. The stirrups with a 6 mm diameter are located in shear-bending regions; likewise, the auxiliary steel bar with a 6 mm diameter is located in compressive regions of concrete; detailed information about those FE models is listed in Table 5. Take specimen SFVF1-@50 as an example (see Figure 16). Φ6@50 denotes the stirrups with a diameter of 6 mm and a distance of 50 mm, whereas SFVF1 means the steel fiber volume fraction is 1%. The load-displacement curves of the FE results are illustrated in Figure 17.

**Table 5.** Information on FE models with stirrup ratios.

| Specimen IDs | SFVF (%) | Shear Span Ratio (λ) | Longitudinal Reinforcements | Stirrup | Stirrup Ratio (%) |
|---|---|---|---|---|---|
| SFVF1-@50 | 1.0% | 2.2 | 2Φ20 | Φ6@50 | 1.88% |
| SFVF1-@100 | 1.0% | 2.2 | 2Φ20 | Φ6@100 | 0.94% |
| SFVF1-@150 | 1.0% | 2.2 | 2Φ20 | Φ6@150 | 0.62% |
| SFVF2-@50 | 2.0% | 2.2 | 2Φ20 | Φ6@50 | 1.88% |
| SFVF2-@100 | 2.0% | 2.2 | 2Φ20 | Φ6@100 | 0.94% |
| SFVF2-@150 | 2.0% | 2.2 | 2Φ20 | Φ6@150 | 0.62% |
| SFVF2.5-@50 | 2.5% | 2.2 | 2Φ20 | Φ6@50 | 1.88% |
| SFVF2.5-@100 | 2.5% | 2.2 | 2Φ20 | Φ6@100 | 0.94% |
| SFVF2.5-@150 | 2.5% | 2.2 | 2Φ20 | Φ6@150 | 0.62% |
| SFVF3-@50 | 3.0% | 2.2 | 2Φ20 | Φ6@50 | 1.88% |
| SFVF3-@100 | 3.0% | 2.2 | 2Φ20 | Φ6@100 | 0.94% |
| SFVF3-@150 | 3.0% | 2.2 | 2Φ20 | Φ6@150 | 0.62% |

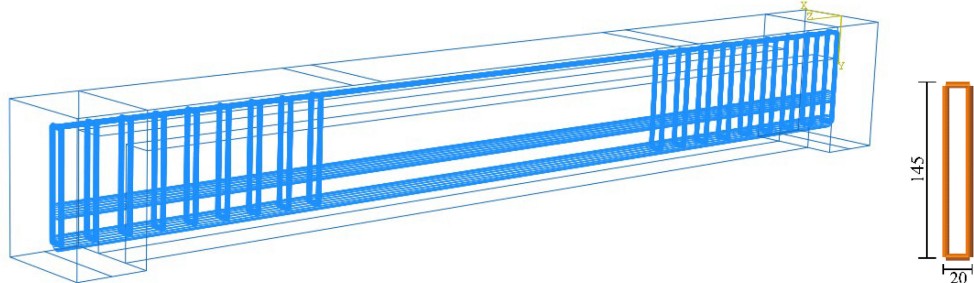

**Figure 16.** Diagrammatic sketch of Φ6@50.

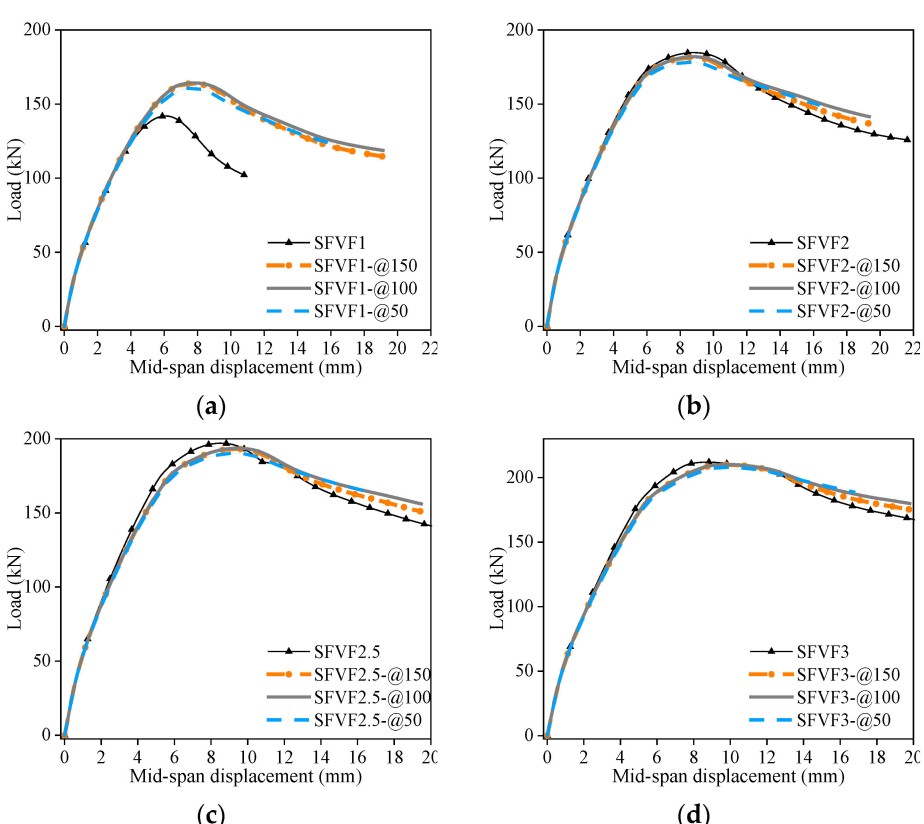

**Figure 17.** Load-mid-span displacements curves: (**a**) SFVF = 1%; (**b**) SFVF = 2%; (**c**) SFVF = 2.5%; and (**d**) SFVF = 3%.

As indicated in Figure 17, for the specimen of SFVF $\geq$ 2%, obviously, with the increase of the stirrup ratio $\rho_{sv}$, the ultimate load $P_u$ has hardly increased (see Figure 17b–d). However, the ultimate load of specimens with SFVF = 1% was found to improve from 141.7 kN to 163.8 kN as the stirrup ratio increased from 0% to 0.62%, which indicated that $\rho_{sv}$ = 0.62% can significantly improve the ultimate capacity and ductility when SFVF = 1% (see Figure 17a). To sum up, the stirrup ratio $\rho_{sv}$ has a significant effect on the shear performance of structures with SFVF $\leq$ 1%, while it has less effect with SFVF $\geq$ 2%.

## 5. Conclusions

In this paper, five specimens were designed and tested to investigate the shear and bending behaviors of UHPC T-section beams, which consider SFVF. Additionally, FEA was carried out, which has been validated by experimental data. Parametric sensitivity analyses were carried out, considering the effect of shear span ratio, reinforcement ratio, and stirrup ratio. The following conclusions can be drawn:

(1) As indicated from the tests, compared with specimen T-1, the ultimate load of T-2~5 increased by 149.7%, 220.5%, 240.2%, and 268.2%, respectively. Thus, the ultimate load of the specimen improves with the increase in steel fiber content.

(2) From experimental observation, SFVF has a big impact on the failure mode of specimens. In detail, the specimens fail in shear failure when SFVF is lower than 2%, while failing in bending failure. Furthermore, as indicated by the changes in failure mode and ultimate loads, the flexural and shear capacities of the beam were enhanced with the increase in fiber content, but the increase in shear capacity was more obvious.

(3) Compared with test data, AFGC code is conservative in the prediction of capacity, which can guide the design of UHPC structures well.

(4) The average value (Av) and standard deviation (SD) of the ratio of ultimate load between FE models and test were 1.020 and 0.063, respectively, indicating that the established FE models in Abaqus (2020) based on the CDP model are effective to simulate the shear and bending behavior of UHPC T-section beams, particularly from the perspective of ultimate capacity prediction.

(5) From the parametric analysis of FEA, with the increase of $\lambda$, the stiffness $K_0$ of the beam in the elastic phase and the bearing capacity $P_u$ of the specimen gradually decreased, but the ductility enhanced, and the failure mode changed from shear failure to bend-shear failure, particularly in specimens of SFVF $\geq$ 2.5%. Additionally, with the increase in the longitudinal reinforcement ratio, the ultimate capacity $P_u$ and initial stiffness $K_0$ of the beam increase, but the ductility decreases, and the failure mode tends to brittle shear damage. Moreover, the stirrup ratio $\rho_{sv}$ significantly affects the shear performance of structures with SFVF $\leq$ 1%, while it has less effect with SFVF $\geq$ 2%.

**Author Contributions:** Conceptualization, J.L.; Software, J.L. and J.Y.; Validation, J.Y.; Resources, J.L.; Writing—original draft, J.L.; Writing—review & editing, J.Y.; Funding acquisition, Y.Y. All authors have read and agreed to the published version of the manuscript.

**Funding:** This work was funded by the Anhui Province Natural Science Foundation of China, grant number 2208085ME151.

**Informed Consent Statement:** Not applicable.

**Data Availability Statement:** The data presented in this study are available upon request.

**Acknowledgments:** The author would like to thank the Anhui Province Natural Science Foundation of China.

**Conflicts of Interest:** The authors declare no conflict of interest.

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
