# Peer review of "Experimental and Numerical Study on the Mechanical Performance of Ultra-High-Performance Concrete T-Section Beams"

_sustainability, doi:10.3390/su15129849_

Round 1

Reviewer 1 Report

Journal name: Sustainability

Manuscript number: sustainability-2438411-peer-review-v1

Full Title: “Experimental and Numerical Study on the Mechanical Performance of Ultra-high Performance Concrete T-section Beam”

Authors: Jianluan Li, Yonggao Yin, Jing Yan

This paper focuses on structural performance of Ultra-High Performance Concrete T-section beams. Both experimental and numerical are developed and the relevant results are presented and discussed.

The paper is well written and formatted; the depiction of the procedure is sufficient. The resolution of the figures is satisfactory and the bibliography is adequate. The research is of interest from an engineering standpoint and, as far as the knowledge of the Reviewer, the results are new and here published for the first time. Nevertheless, the following questions need to be addressed by the Authors prior to publication.

11)      Title: replace “beam” with “beams”.

22)      English language is correct, but a careful check is suggested to make it more fluent.

33)      Abstract: lines 14-24 are conclusions; please focus on the main contributions of the manuscript.

44)      Introduction: it is unclear to the Reviewer why the Authors refer to “bridges” instead of “beams” (lines: 31, 65, 67 and 74).

Authors are invited to submit a rebuttal version of the manuscript; the Reviewer’s decision is “Minor Revision”.

Please see comment #3.

Reviewer 2 Report

The authors used UHPC in T-beam. They conducted both experiments and numerical research. The paper is generally good but it needs improvement. Followings should be carried out before acceptance:

The abstract should contain important results of the study.

Line 42 where the references 11, 12 ,14 and 15. It seems that reference numbering is wrong

Intoduction section is short and should be improved.

The number of total reference is very low. It should be significanlty increased.

Novelty is not clear. Very same studies are already exists. What is the difference?

Which type of steel fiber was utilized? Add photos

The authors can add a paragph to introduce steel fibers. They can mention types of steel fibers and etc. The following studies can be utilized for this purpose: improvement in bending performance of reinforced concrete beams produced with waste lathe scraps; performance assessment of fiber-reinforced concrete produced with waste lathe fibers; performance evaluation of fiber-reinforced concretes produced with steel fibers extracted from waste tire; investigation on improvement in shear performance of reinforced-concrete beams produced with recycled steel wires from waste tires; geopolymer concrete with high strength, workability and setting time using recycled steel wires and basalt powder

Change Fig 2b. There is no need a naked man appear in photo.

Quality of fig 3 should be improved.

No need for Fig 5.

Line 183. concrete damage plasticity (CDP) material model [19, 27-29] in Abaqus was. The studies are irrelevant with abaqus and CDP. These references can be replaced with following which includes abaqus with cdp: bending performance of dapped-end beams having web opening: Experimental and numerical investigation ; experimental and numerical investigation of shear strength at dapped end beams having different shear span and recess corner length ; experimental and numerical investigations of steel fiber reinforced concrete dapped-end purlins ; damages on prefabricated concrete dapped-end purlins due to snow loads and a novel reinforcement detail ; numerical evaluation of effects of shear span, stirrup spacing and angle of stirrup on reinforced concrete beam behaviour

Fig 8 is not good. It shoud be changed. You can refer prevous suggested papers.

Add recent studies on this subject to introduction. There are many studies on the introduction for this topic.

Conclusion should be improved. The recommendation consdiering all test should be given for engineers.

Reviewer 3 Report

The referred manuscript uses experimental and numerical investigation to study the mechanical behaviors of UHPC beams with the aim of understanding its bear capacity when dense and randomly-distributed steel fiber is added. Overall, the structure of the paper is well organized, the study method and result are clearly presented, and the topic and research findings are valuable in the engineering field.

However, there are a few concerns about the investigation method/result and paper writing. They should be fixed before being accepted to the Journal of Sustainability.

Major concerns:

1)     The Coherence of the wording flow is a big concern in this paper. Transition phrases are rarely or not properly used in the manuscript. Furthermore, the breaks of the sentences are messy. Commas and period symbols are not properly used in many paragraphs. For example, in Abstract, on line 9, the comma after “(SFVF) ” should be replaced by a period. On line 17, “otherwise” is not properly used. Similar problems can be found in the other sections, and they are not enumerated here. These issues will significantly deteriorate the paper's readability. Actually, I have to read them multiple times to understand the author’s statements. Careful proofreading needs to be completed by the authors.

2)     Many expressions do not follow English or engineering convention. For example, what is “floral toughness?” On line 99, “configuration” is not an appropriate word. On line 92, is “horizontally ” the correct word here? On line l02, “diagram” is not the appropriate word. “Between … with …” is not a correct phrase. On line 158, the “effective area of fiber effect” is quite confusing. on line 181, “analytical” is not the correct word here.   “FEM” is a mathematical method but not an engineering analysis approach. It should be replaced by “FEA.” on line 108, “specimen occurs as a shear failure” is a confusing statement. On line 252, “modality” is not appropriate. On line 259, “cloud” should be “contour.” The similar issue can also be found in many other places in the paper.

3)     The last paragraph of the Introduction should be a recap of the paper. However, the parametric analysis is not included. Furth more, the first sentence in this paragraph should be either moved to the conclusion or beginning of the Introduction, or directly deleted. It breaks the transition between the previous paragraph and the subsequent discussions.

4)     “bending failure,” “bending damage,” “shear failure,” and “shear damage” should be clearly defined in the paper. They are easily misunderstood as model I and II damage.

5)     Figure 2 is not completed and is not self-explanatory. The notes should be given to indicate strain gauge symbols.

6)     Steel fiber ratio is based on volume or mass? How about the shear span ratio, reinforcement ratio, and stirrup ratio?

7)     Only testing one specimen for each SFVF ratio seems less persuasive. The steel fiber is randomly distributed, and the failure mode may also vary in each specimen with the same SFVF.

8)     On line 108 of page 3, “T-4 and T5 are shear damage after bending yielding” is confusing. What is “bending yield?”

9)     On line 111 of page 3, it is unclear how to derive the conclusion, “the flexural and shear capacity of the beam enhanced with the increase of fiber content, but the increase of shear capacity is more obvious.” I don’t think we can draw such a conclusion based on the failure mode change.

10)  Similar to 7), on line 121, how to draw the conclusion, “And the increase in steel fiber content is more advantageous in improving the shear capacity of the beam than the flexural capacity.” From Figure 6, the bending resistance increases obviously with the addition of steel fibers.

11)  It seems that Figure 5 is not used in the paper. Why not delete it?

12)  On line 134, “there is no obvious tensile or compressive partitioning” is not clear. Also, on line 135, why do the authors say “a clear assumption of flat cross-section is exhibited?”

13)  In Figure 6(b), the strain at the bottom strain gauge should be larger than the upper gauge, because it locates at the bottom of the beam. It is not clear why the strain is smaller than the upper one.

14)  In the FEA analysis, CDP model is used. However, this mode is designed for general concrete. How did authors consider the steel fibers' contribution to UHPC material property?

15)  On line 185 and 186, it states that Equation (7) is a fitting formula for UHPC but Equation (7) is noted as for longitudinal reinforcement.

16)  From line 189 to line 195, the authors discuss the argument to pick the stress-displacement model to define tensile behaviors after failure. However, in Abaqus, it seems that this material model is not provided. How did the authors implement this material model?

17)  On line 197, it states that Equation (8) describes the tensile properties of UHPC but Equation (8) is noted as a compression model.

18)  Equations (7), (8), and (10) are not introduced in the manuscript. They should be carefully discussed.

19)  On line 209, it states “the end of tensile strain hardening.” Is there tensile strain hardening in CDP model?  

20)  How do damage coefficients degrade the concrete stiffness (Equations (10) and (11))? It is not discussed in the manuscript.

21)  On line 216, the index of stiffness degradation is selected as 1 based on parametric analysis. The author should discuss how to use parametric analysis to decide the index of stiffness.

22)  In Figure 9 (a), why did the author use perfect plasticity?

23)  Figure 9 (b) is not a stress-strain relation.

24)  Was the FEA analysis completed in Abaqus/Standard or Explicit?

25)  In the FEA analysis, mesh convergence should be performed before compromising model accuracy. Without knowing if the model converges has been achieved, scarifying the accuracy for computation efficiency is not a wise decision.

26)  On line 241, it states the bond-slip is not considered. In Abaqus CDP model, this problem can be modeled by tension stiffening. Authors may consider improving the simulation result with these technics in Abaqus.

27)   The comparison of damage mode between FEA and experiments is not clear. Figure 12 should be reproduced.

28)   On line 293, the reader may be confused with the statement, “the failure mode tends to brittle shear damage,” because there is no image showing the failure mode change.

29)  On line 295, the conclusion, “the best design value of ρ is 5.6% when SFVF=1%” should be carefully justified.

30)  On line 315, how to find “the ductility of specimens increases” in Figure 16?

Minor:

1)     Using symbols to identify different curves is highly recommended because readers are difficult to differentiate the plots when printing the paper in black-white colors.

2)     Some phrases or sentence seems quite reductant or twisted. They should be either deleted or rephrased. For example, on line 50, “to investigate the size effect of beams” should be deleted. On line 116, “The ultimate capacity of T-1 (no steel fibers) compared with that of T-2~5 (with steel fibers) shows” is not appropriate. The subject should be a comparison but not a specimen. On line 47, “numerous studies [6,17–20,26] have designed a series of tests” is not appropriate. The study cannot design tests. These issues can be found in many other places and are not listed here. They should be carefully proofread.

3)     There are many editorial errors and typos. For example, on line 124, “derived” may be replaced by “deviated.” On line e146, “with” should replace by “and.” On line 170, “subsequence” should be replaced by “subsequently.” On lines 305 and 306, “local” should be replaced by “located.” Particularly, the symbols used in Equation (1) ~ (11) should be carefully proofread. Some formats and shapes are not consistent. For example, on line 163, what is V_RD,u? 

The manuscript shows a high quality of English writing. For example, there are almost no severe grammar errors in the paper. However, many terms and phrases do not follow English conventions. When picking up words, they should be carefully compared. 

Another issue is the wording flow. Appropriate transitions are missing in the paper. 

Reviewer 4 Report

The paper presents a well designed, comprehensive experimental and numerical research on the performance of UHPC with steel fibers. The results are interesting, and the methodology is quite sound and it was properly followed. The FEM sensitiveness study is of particular interest.The conclusions are interesting and they are coherent with the main findings of the research. 

Regarding section 2.3 on design codes with respect to experimental results, the reviewer considers that this section could be improved by means of further insight on the results. It should be noted that the code design equations include safety factors that are supposed to cover different uncertainties. Therefore, and interestingly enough, the shear failure from the code seems quite accurate, or it could even be unconservative if material safety factors disappeared, while the bending failure still seems too conservative. This could be used by the authors to complete the section by providing further discussion (actual safety margins, etc.) on the design equations.

The paper is easy to follow but there are several grammatical flaws that require revision of the English language by the authors, mostly concerned with the choice of prepositions. 

Round 2

Reviewer 2 Report

The paper can be accepted

Author Response

Thank you very much for your comments and suggestions and thank you for recommending our paper for publication in sustainability.

Reviewer 3 Report

The revised manuscript shows significant improvement. The authors completed a decent job and the paper is almost ready for publication. However, after carefully reviewing the revised version, I noticed that there are multiple issues that were listed in the original review report but were not completely addressed. These problems are critical for published academic documents. The authors should carefully resolve the issues listed below.

1) Most wording flow coherence issues mentioned in the original review report have been fixed but the proofreading was not completed thoroughly. I noticed the following problems:

a. In the abstract, the transition words “then” on lines 9 and line 12 can be replaced. 

b. From line 32 to line 36, the long sentence should be properly broken into short sentences. For example, on line 33, the comma after “it” can be changed to a period. Similarly, on line 72, the comma before “especially,” on line 119, the comma before “load,” on line 101, the comma before “the,” on line 133, the comma before “the,” on line 182, the comma before “of,” on line 185, the comma before “then,” on line 346, comma before “take” should be changed to period. 

2) Similarly, in the revised manuscript, I also noticed multiple typos/grammar/editorial problems as mentioned in the original review report. They are listed below. 

a. “aim to” and “aiming to” are mixed in the paper. The format should be consistent.

b. On line 37, “influencing” seems reductant. 

c. On line 47, the comma after “beams” is reductant. 

d. On line 54, “to calculate” may change to “when calculating.”

e. The usage of prepositions is a concern. For example, on line 46, “by” may change to “in.” on line 353, “from” may change to “in.”

f. On line 66, does “impact load” refer to dynamic load?

g. On line 68. “although” is not proper here.

h. On line 72, “evaluate” should be in a passive tone.

i. On line 82, “bending shear” may change to “bend shear” to maintain a consistent format.

j. On line 97, “subsequence” is not correct.

k. Online 96, the sentence “the bridging mechanism ….” seems irrelevant to the context. 

l. On line 141, “for specimens in this study, due to a longitudinal reinforcement ratio of 7% and a stirrup ratio of 0%” is confusing.

m. On line 155, “and” is missing after “SFVF,”.

n. On line 177, again, it is not clear what “fiber effect” is.

o. On line 191, is “while” a proper transition word?

p. On line 194, “has” may change to “having.”

q. On line 227 and 228, “in contrast” and “in summary” are not appropriate phrases. 

r. On line 150, “is” is missing after “which.”

s. The author only changed “FEM” to “FEA” on line 227. However, “FEM” is still massively used all along the paper.

t. The author only changed “local” to “located” on line 345. There is no change on line 344.

u. On line 341, “affecting” is not correct.

v. On line 342, “1” is a typo.

w. On line 372, “phenomenon” may change to “observation.”

x. On line 381, “efficient” may change to “effective.”

3) On line 290, it states, “taking specimen T-0,” but in Figure 12, the specimen is T-1. 

4) From line 314 to line 316, the author discusses the failure mode change due to the shear span ratio effects based on Figure 13. However, without showing FEA model failure mode, it is difficult to say “and the failure mode changed from shear failure to bending failure” on line 316.

5) In Figure 5(a), the authors accepted the review suggestion and added symbols to each curve. However, the author did not address the same confusion in Figures 9, 13, 14, and 16. 

6) On line 334, it states that, based on Figure 13(a), “the best design value of ρ is 5.6% when SFVF=1%.” It is not clear how to draw this conclusion based on Figure 13 (a).

The transition phrases should be widely used in the paper. Also, the usage of prepositions should be improved. 

Reviewer 4 Report

The authors properly amended the issues detected in the original version of the paper.

Author Response

(The authors gave the same response as above.)

Round 3

Reviewer 3 Report

All of my concerns have been addressed. The paper can be accepted for publication.